# Integrating Information from Natural Language Parse Tree to Code Generation

**Hung Phan & Ali Jannesari**
Department of Computer Science, Iowa State University, USA
{hungphd, jannesari}@iastate.edu

## Abstract

While more and more research works have considered Natural Language artifacts as the inputs of software engineering (SE) research, such as code generation, information about their graph/tree representations needs to be carefully considered. In this work, we propose an approach for integrating information on NLPT on multiple problems in SE tasks. The preliminary experiment shows that augmenting information of NLPT can improve the code generation from pseudocode.

## 1 Introduction

Neural Machine Translation (NMT) is an important module that was used in different approaches of pseudocode-to-code translation proposed in the work of Kulal et al. Kulal et al. (2019) and Zhang et al. Zhong et al. (2020) OpenNMT Klein et al. (2017) is a well-known implementation for NMT. OpenNMT considered the input pseudocode in its most trivial representation: sequence of words. In Natural Language Processing (NLP), Natural Language Parse Tree (NLPT) is a well-known tree representation of natural language artifacts. For Software Engineering (SE) tasks in general and pseudocode-to-code translation specifically, information about NLPT of pseudocode is omitted. In this work, we propose NLPTAnalyzer, an engine for integrating and evaluating the effectiveness of augmenting information of NLPT to existing SE models for code generation by different augmentation strategies. The replication package is available at here[1].

## 2 Approach

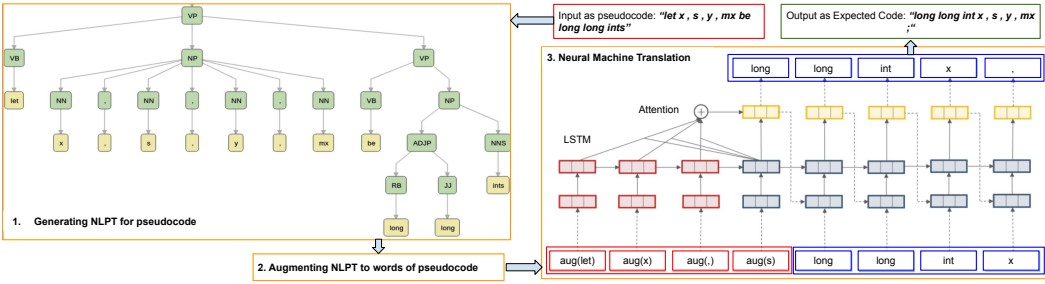

Figure 1: Overview Architecture of NLPTAnalyzer in Pseudocode to Code Translation

Overview Architecture of NLPTAnalyzer can be shown in Figure 1. Given the input query as pseudocode, the NLPT of a pseudocode will be generated by StanfordCoreNLP Manning et al. (2014). Next, we extract information of ancestor nodes for each leaf node of NLPT for augmentation. We implement two strategies of augmentation: $Aug - 1$ and $Aug - 2$. For each leaf node, the $Aug - 1$ strategy integrates the value of the leaf node and the value of its closest ancestor. In $Aug - 2$ strategy, information of the closest ancestor of the closest ancestor node for a leaf node is included. Next, the OpenNMT model is used to train on the information of pseudocode augmented with NLPT and the source code.

---

[1] https://tinyurl.com/46buxd62

| Strrategy | BLEU-4 | Input (first 3 words) | Translated Result |
|---|---|---|---|
| Original | **48.53%** | let x , s | **long long int x , s , y ;** |
| Aug-1 | **48.94%** | VB let NN x , , NN s | **long long int x , y , s , mx ;** |
| Aug-2 | **48.82%** | VP let NP x NP , NP s | **long long int x , s , y , mx ;** |

Table 1: Comparison of NLPTAnalyzer with two augmented strategies versus original text

## 3 EXPERIMENT

**Configuration.** We use the SPOC dataset for training the pseudocode-to-code translation, with **14960** programs for training, **1815** programs for validation and **1749** programs for testing. We use the default configuration provided by OpenNMT Klein et al. (2017) to train our model, with **10000** steps for training, with numbers of layers for encoder and decoder as **2** for each. We use the $hidden_states$ as **500** for each training layer. We use a popular metric BLEU-4 cumulative Klein et al. (2017), to compare the translated result and the expected result.

**Results.** The experiment result is shown in Table 1. We see that with the new input as textual information of pseudocode and its context, the BLEU-4 slightly increases from 0.3% to 0.4% for each augmented strategy. Another observation is that not only the closest ancestor node of a leaf node of NLPT brings the information to improve the accuracy, but the ancestor from a higher level of NLPT can also contribute to the accuracy of pseudocode-to-code translation. Examples of translated results proposed by different augmented strategies can be shown in the last column of Table 1. While the result trained on the original pseudocode missed the information of variable "$mx$", the augmented strategy can include the role of "$mx$" in pseudocode as an element in a noun phrase (NP) and Noun (NN) tags to suggest the NMT model learned the output correctly.

## 4 DIRECTION ON IMPROVING NATURAL LANGUAGE PARSE TREE FOR SOFTWARE ENGINEERING TASKS

Information extracted from NLPT hasn't reflected SE properties, such as what is the respective mapping of a word in pseudocode to its implementation. For example, while "$x$" and "$ints$" words are both tagged as Noun, further classification is required since "$x$" is a variable and "$ints$" is the data type. Besides, from the above example, we see that there are two important types of semantic meanings that words can have in a query. First, some words appeared in the implementation, such as words about variables and data types. Second, some words didn't appear in the implementation but can be mapped with types of statements. For example, the word "$let$" can be mapped to a sign of declaration statement in the implementation, or "$be$" can be mapped to the equal sign. In future works, NLPT should contain this information to support tasks for code inference.

## 5 RELATED WORK

From our knowledge, SPOC dataset Kulal et al. (2019) is the latest dataset for pseudocode-to-code generation. This dataset contains source code and corresponding pseudocode made by Amazon Turk Workers. From this dataset, Kulal et al. implemented an approach for integrating OpenNMT pseudocode-to-code translation with a program analysis module to re-rank the best-translated candidates per each line of pseudocode. Zhong et al. (2020) improved the work from Kulal et al. by introducing another module for eliminating trivial pseudocode and fixing the translated results from OpenNMT by proposing templates of errors and implementing solutions for each template.

## 6 CONCLUSION

In this work, we propose NLPTAnalyzer, a tool for integrating information of NLPT to the input as NL artifacts of SE problems. We show that with augmented information, NLPT can improve the accuracy of the pseudocode-to-code translation. Future works should integrate SE-based properties to NLPT and apply the augmentation for other code generation techniques, such as code search by embedding Guo et al. (2020).

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

## A APPENDIX

### A.1 UNDERREPRESENTED MINORITY (URM) CRITERIA

The authors of this paper satisfy the URM criteria.

