# OpenReview forum: "INTEGRATING INFORMATION FROM NATURAL LANGUAGE PARSE TREE TO CODE GENERATION"
_ICLR.cc/2023/TinyPapers — Submitted to Tiny Papers @ ICLR 2023_

### Official Review · Reviewer_s17k · 2023-03-27

**Confidence:** 3

**Summary Of Contributions:**

An interesting paper but lacks novelty.

**Rating:**

Great Start (GS): a submission which meets some of the reviewing criteria but has room for improvement

**Strengths And Weaknesses:**

This paper proposes an approach for integrating natural language processing techniques (NLPT) to solve multiple problems in software engineering tasks. The preliminary experiment shows that augmenting NLPT information can improve the code generation from pseudocode.

Strengths:

1.The proposed approach is simple and effective in improving code generation from pseudocode.

Weaknesses:

1.The proposed approach may be considered somewhat incremental, and the method introduction is a bit vague which makes it hard to understand.

2.The performance improvement is relatively small.


**Suggested Changes:**

To strengthen the approach, more comprehensive experiments and stronger technical improvements are needed.

---

### Official Review · Reviewer_3MpW · 2023-04-01

**Confidence:** 4

**Summary Of Contributions:**

In this submission, the authors describe NLPTAnalyzer, an approach to translating pseudo-code into code. The main idea is to take the Natural Language Parse Tree (NLPT), a tree representation of natural language artifacts, into account. They train an LSTM-based architecture (including attention) and report initial experimental results.

**Rating:**

Great Start (GS): a submission which meets some of the reviewing criteria but has room for improvement

**Strengths And Weaknesses:**

Strengths:
- Encoding the tree structure of coding problems into sequential learning approaches is important and should be made more prominent
- Their approach shows a slight increase in performance

Weaknesses:
- The architecture decision seems outdated (with Transformers) + tree positional encodings available that are known to improve the performance on such tasks.
- The gain is not significant, it is not clear from the experiments how high the variance is.

**Suggested Changes:**

I would recommend the authors to compare to Transformers + tree positional encoding on this problem. Additionally, I would love to see the variance of the experimental results.

---

### Official Review · Reviewer_YK5Z · 2023-04-02

**Confidence:** 4

**Summary Of Contributions:**

The paper demonstrates the effectiveness of augmenting information of natural language parse tree to existing SE models pseudocode-to-code translation using 2 different augmentation strategies : ancestor of leaf node and closest ancestor of the closest ancestor of the leaf node.

**Rating:**

Great Start (GS): a submission which meets some of the reviewing criteria but has room for improvement

**Strengths And Weaknesses:**

## Strengths
1. The idea of augmenting pseudocode with parse tree information is inspiring and worth exploring further for other code generation tasks
2. The motivation is clear and the methodology is sensible, the figure is self explanatory
3. Section 4 is well written and crucial to positioning the usefulness and impact of the paper and its future directions


## Weaknesses
1. The code was not easy to understand or reproduce at the link attached
2. Table 1 has a column 'Input(first 3 words)' while the 'Translated Result' has the code for the entire pseudocode statement I suppose, it is a bit unclear as to how the 'mx' appears in Aug-1 and Aug-2 strategies from just the First 3 input words. If the entire pseudocode statement was input and truncated at 3 words for the sake of fitting in the table then the results make sense and would suggest the authors to add this note in the caption for better understanding
4. The intuition behind the Aug-1 and Aug-2 strategies are missing.
5. Very marginal improvement in BLEU-4 score ~0.4% for 1749 testing examples

**Suggested Changes:**

## Suggested Changes
1. To fully understand the effectiveness of this methodology I suggest the authors evaluate the functional correctness of the generate code to better evaluate the effectiveness of the proposed strategies. BLEU scores are a good measure of the surface-level correctness of a translation however, metrics like success rate as shown in Kulal et al. 2019 will be helpful in evaluating the quality and correctness of the code itself.
2. The authors may move the implementation details from Section 3 to the Appendix to create more space in the main paper
3. Adding the intuition and some motivation around Aug-1 and Aug-2 as well as using parse tree information in Introduction will make this work more impactful
4. The abstract can be modified to sound more natural, it reads very curt and formal

### Minor Typos
1. Citations are not properly formatted in Section 1
2. Typos in Section 3

---

### Meta-Review · Area_Chair_DiG6 · 2023-04-05

**Recommendation:** Invite to archive
**Confidence:** 4

**Metareview:**

**Summary**
* The paper describes NLPTAnalyzer, an approach to translating pseudo-code into code. The main idea is to take the Natural Language Parse Tree (NLPT), a tree representation of natural language artifacts, into account. They train an LSTM-based architecture (including attention) and report initial experimental results.

**Strengths**
* Augmenting pseudocode with parse tree information is an important approach. It’s worth exploring further for other code-generation tasks.

**Weaknesses**
* The gain in BLEU-4 score ~0.4% for 1749 testing examples is not significant.
* The architectural decision seems outdated. As ( Transformers + tree positional encodings) are known to improve the performance on such tasks.
* The attached code wasn’t easy to understand for reproducibility purposes.



**Summary:**

The paper describes NLPTAnalyzer, an approach to translating pseudo-code into code. The main idea is to take the Natural Language Parse Tree (NLPT), a tree representation of natural language artifacts, into account. Main strength, augmenting pseudocode with parse tree information is an important approach. It’s worth exploring further for other code-generation tasks. Main weakness, the reported gain in BLEU-4 score ~0.4% for 1749 testing examples is not significant

**Reason For Not Giving A Higher Recommendation:**

* The architecture used is somehow outdated and using new methods that have shown to perform well in such tasks would be great.
* The gain reported in the result by using the method in small data is not significant.

Hence, revising the work by using recent architectures and comparing the results would be interesting.


**Reason For Not Giving A Lower Recommendation:**

N/A

---

### Decision · Program_Chairs · 2023-04-10

Invite to archive